# Towards a Linked Open Code

Ahmed El Amine Djebri[1][0000−0003−2917−5085], Antonia
Ettorre[1][0000−0003−4868−2584], and Johann Mortara[2][0000−0002−1779−5511]✉

[1] Université Côte d'Azur, Inria, CNRS, I3S, France
[2] Université Côte d'Azur, CNRS, I3S, France
{ahmed-elamine.djebri,antonia.ettorre,johann.mortara}@univ-cotedazur.fr

**Abstract.** In the last two decades, the Linked Open Data paradigm
has been experiencing exponential growth. Regularly, new datasets and
ontologies are made publicly available, and novel projects are initiated
to stimulate their continuous development and reuse, pushing more and
more actors to adhere to the Semantic Web principles. The guidelines
provided by the Semantic Web community allow to (*i*) homogeneously
represent, (*ii*) uniquely identify, and (*iii*) uniformly reference any piece
of information. However, the same standards do not allow defining and
referencing the methods to exploit it: functions, procedures, algorithms,
and code in general, are left out of this interconnected world. In this
paper, we present our vision for a Web with *Linked Open Code* in which
functions could be accessed and used as Linked Data, allowing logic har-
nessing the latter to be semantically described and *FAIR*-ly accessible.
Hereafter, we describe the challenges presented by the implementation of
our vision. We propose first insights on how to concretize it, and we pro-
vide a non-exhaustive list of communities that could benefit from such
an ideal.

**Keywords:** Semantic Web · Ontologies · Feature Identification · Linked
Data · Linked Open Code

## 1 Introduction

The Web is growing stronger semantically. More ready-to-consume data, ser-
vices, and AI-based systems relying on Semantic Web are regularly published.
We witness the emergence of the Semantic Web in different unrelated fields such
as AI, IoT, networking, medicine, or biology. Within each, papers are being
published, wikis are being created, and code is made available. All these dif-
ferent fields share their data through a unique structure, reaching the vision of
Tim Berners-Lee who mentioned: *"Semantic Web promotes this synergy: even
agents that were not expressly designed to work together can transfer data among
themselves when the data come with semantics."* [1].

While Semantic Web offers ways to store metadata to reuse them seman-
tically, code is not used on the Semantic Web to its full potential. Hence, the
problem we seek to tackle is: *how to take advantage of code as a pre-existing,
structured, and functional type of data in Semantic Web?*

Code for data manipulation is actually either (*i*) not needed for simple operations as existing standards offer sufficient functionalities (*e.g.* functions in SPARQL [2]) or (*ii*) used at a higher level in the Semantic Web stack, where users download and build code from open repositories provided to them by the data provider. However, these two approaches exhibit some limitations: in (*i*), the capacities of SPARQL functions are limited and in (*ii*), despite the availability of the code on the Web, the possibility to have a link between the semantics of data and the semantics of code is not fully harnessed. We believe that code should be treated as a special type of data. The use of functions or methods on Semantic Web is usually studied for limited use-cases, such as schema validation (*i.e.* `sh:JSFunction` representing JavaScript functions to be used in *SHACL* engines). We think that the link between code and Semantic Web remains superficial. Functions are not semantically shared as and with data.

We argue that functions, as parts of code, are easily referenceable and can be identified by a defined set of metadata. However, defining their semantics is challenging as functions can be seen from different levels of granularity. Finally, although source code can already be browsed and referenced online at multiple levels of granularity by platforms such as *Software Heritage* [3] or GitHub's permalinks, they do not provide any description of the functionality implemented by the code, thus limiting the code reusability.

## 2   Code on Semantic Web

Data published on the Semantic Web are often followed by instructions on how to access, read, manipulate, and query them. Ontologies are documented in scientific literature and wikis, offering insights on their semantics, and tools for data manipulation are being provided. An increasing number of developers give open access to public source code repositories hosted on data providers such as GitHub. Academics can publish code directly alongside their paper [3] for frameworks they developed, encouraged by new policies from editors to foster reproduction and reuse of research results [4].

In contrast with Linked Data, code files are often seen as single documents on the Web as the transition between the *document-based* view and the *data-based* one has not affected them on a fine-grained level. Hence, the link between data and the code artifacts directly involved with it remains limited. We believe that since both resources (data and code artifacts related to them) are available on the Web, an effort should be made to provide code in the same format as and alongside data.

### 2.1   Adapting Code to Semantic Web

According to the *Web Service Modeling Ontology Primer* (WSMO) [4], a function is not only a syntactical entity but also has defined semantics that allows

---

[3] https://blog.arxiv.org/2020/10/08/new-arxivlabs-feature-provides-instant-access-to-code/

[4] https://www.acm.org/publications/policies/artifact-review-and-badging-current

evaluating the function if concrete input values for the parameters are given. However, the structure of functions defined in most programming languages is more complex than in the definition provided by the WSMO as their computation may rely on data other than the values specified as its parameters such as (*i*) results of other functions defined in the same project or an external library, or (*ii*) attributes of an object for object-oriented methods. These values are provided to the functions by their execution environment, as the *Java Virtual Machine* (JVM) for Java-based systems.

For a function to be compliant with our case, it should (*i*) depend on the standard libraries of a language-version, either directly or transitively through other referenceable functions, and (*ii*) not rely on out-of-scope variables. Property (*i*) applies recursively to any function call inside the function itself. If a code is to be written in an inline mode, any other function call within the same function must be replaced by a set of instructions depending only on the standard libraries of a defined language-version. Achieving (*ii*) requires binding out-of-scope variables to their values.

Many challenges arise from this new definition, starting with the fact that the existing code repositories do not provide a "function-based" view. As a consequence, we should figure out how to turn those into referenceable, reusable resources. The following challenges, presented in fig. 1, are to be addressed.

**Referencing functions** Function structure and signature in code make it easily recognizable. The signatures usually contain information such as the function's name and its typed arguments (*cf.* fig. 2). Such information can be represented as linked data while attributing a unique identifier for function definitions.

The idea is to allow Linked Data providers to publish, following the Semantic Web principles, the code of functions, and their metadata. Furthermore, one may include an additional level of granularity to existing IRIs referencing code entities (repositories, folders, files, fragment), helping to reference functions and keep track of their provenance. For example, a code file archived on *Software Heritage* with the IRI `swh:codeFile` helps addressing the function `fn` using the IRI `swh:codeFile_fn_1` (instead of referencing fragments of code with no defined semantics).

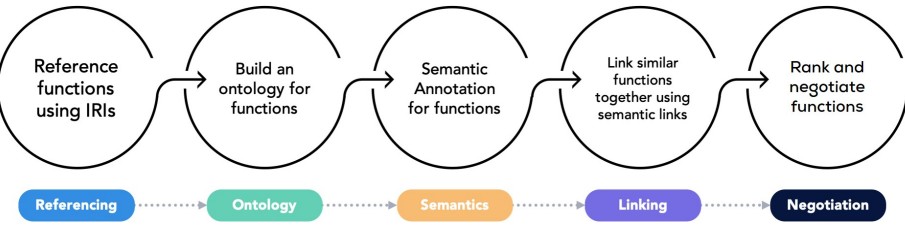

**Fig. 1.** Challenges to achieve a first working prototype of Linked Open Code

**Fig. 2.** Comparison of metadata provided function signature in Python and C++

**An ontology for functions** A crucial step to bring functions to the Semantic Web is the definition of an ontology to represent them. Such ontology must describe four aspects:

1. Versioning: the version of the function, programming language, provenance.
2. Relational: relations between functions (inclusion, dependencies, etc.).
3. Technical: code, arguments, typing, etc.
4. Licensing: although all open source licenses imply free-use and sharing of code [5], some may impose restrictions on the reuse (*e.g.* crediting the original author), hence this information needs to be provided to the user.

**Annotating functions semantically** During this step, the defined functions are mapped each with their signature and feature metadata. An Abstract Syntax Tree (AST) analysis is applied on each to identify the components constituting the signature of the function (name, parameters, ...) that will then be used as values for the properties defined in the ontology. As a result, the user will be able to query the knowledge base to retrieve the function matching the given constraints. In parallel, a feature identification process is executed to identify the functionalities implemented by each function and annotate them accordingly. The whole process is depicted as in fig. 3. Multiple techniques for the identification of features have already been proposed [5] and need to be adapted to our context.

**Linking functions** After having identified the features provided by the functions, we can use this information to semantically link functions fulfilling similar goals. Indeed, two functions being annotated with the same feature can be considered as different implementations for the same functionality as perceived by the user. Therefore, we can link them with standard predicates such as `owl:sameAs, skos:exactMatch, skos:closeMatch` or custom predicates offered by other existing ontologies. Alongside semantics, the dependency must be taken into account to link related functions together. Based on this criterion, functions relying on the results provided by other functions (including the function itself in the case of recursive calls) will be semantically connected.

---

[5] https://opensource.org/licenses

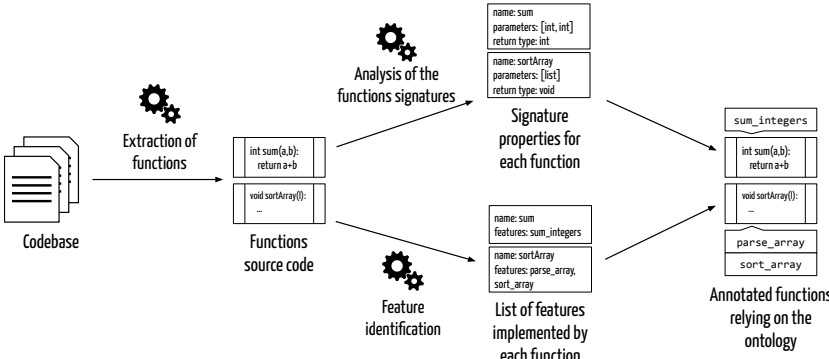

**Fig. 3.** Overview of the process for semantic annotation of functions

**Ranking functions** The same functionality can be implemented in different ways and using different programming languages. To provide the most efficient implementation, there is a need to rank functions according to several parameters. One example can be the feedback of the community, as a repository where usage statistics for functions are being kept for ranking purposes, alongside other information such as the number of times a function was starred, forked, or upvoted by users. It is also possible to signal issues related to security flaws. Performance evaluation can also be used as a ranking criterion. A Semantic Web Engine like *Corese* [6], coded in *Java*, would use functionality implemented in *Java*. However, the same functionality, implemented in *Python*, can deliver better performance for the same tool if used with a *Python* wrapper. This aspect is meant to link code with experience. We can imagine users sharing their execution log, which may contain elements about hardware specification, operating system, language version, etc.

**Negotiating functions** Users may take advantage of the implemented content negotiation to get suitable function definitions for their use-cases. This is done by using HTTP headers, or non-HTTP methods like Query String Arguments (*QSA*). Users negotiate functions that suit their current environment to access and manipulate Linked Data. For instance, a user working with *Corese* may send a request to the function catalog, asking for the *Java* implementation of functions alongside their query for data. Negotiation can rely on the previous step, by proposing the best function to the users according to their specifications.

The realization of this vision would be a framework through which the user would use SPARQL to query a catalog of functions (section 2.1) for the implementations of needed functionalities meeting architectural and user-defined requirements. The fetched code artifacts can then be composed to build a tailored software system. However, the automatic composition of software artifacts

---

[6] https://github.com/Wimmics/corese

is a whole challenge in itself [6] and is out of the scope of this vision. Concretizing the vision raises other challenges (*e.g.* scalability) that will need to be addressed when designing the actual solution.

## 2.2   First approaches towards Linked Open Code

The scientific community started taking promising steps to tackle the aforementioned points and make code semantically and uniformly accessible on the Web.

Initial works such as [7,8,9] focused on remote execution, through SPARQL queries, of code explicitly written for the Semantic Web. While [7] and [8] deal with SPARQL functions, [9] defines a new scripting language, *LDScript*, but its expressiveness is limited when compared to conventional programming languages. However, none of these approaches enables users to discover, download, and locally execute the best implementation of a given functionality in a required programming language.

More recent works aim to make code written in any language uniformly accessible through semantic queries. Ontologies are defined to describe code either for a specific language, like the $R$ ontology [10]; a specific paradigm, such as object-oriented languages with CodeOntology [11]; or independently of the used technology as done by [12]. While [10] does not discuss the link between functions and data and lacks a way to capture the semantics of the functions, [11] and [12] have been extended respectively in [13] and  [14,15,16,17] to tackle these limitations.

The work presented in [13] relies on CodeOntology for the implementation of a query answering system. The user's queries are translated into SPARQL queries and evaluated against a repository containing the RDF definitions of functions. Those functions are discovered and annotated using CodeOntology to describe their structure and DBpedia for semantics. Though this approach is similar to our vision for what concerns the discovery and semantic annotations of the functions, it differs as it remotely executes functions to answer the user's query while our goal is to find and return the best implementation of the requested functionality. Moreover, we aim to be able to deal with every kind of function despite the paradigm of the language in which they are implemented.

In [14,15,16,17], De Meester *et al.* broaden the vision presented in [12] by introducing new concepts, *e.g.* content negotiation. These approaches are very similar to our vision, with the main difference (which is also one of the main challenges of our approach) that we aim to automatically discover, identify and annotate the source code, while these previous works foresee the manual publication of description and implementations by developers. The works discuss briefly ranking the functions, but do not mention what metrics are to be used.

The last very recent initiative is Wikilambda [18] by the Wikimedia foundation. Its aim is to abstract Wikipedia by offering several implementations of functions allowing, firstly, to render the content of Wikidata in natural language using predefined templates and, as a final goal, to make the referenceable functions available on the Web. The main limitation of such an initiative is that the repository needs to be manually populated with functions written by the

community, meaning that the success of the approach depends on the expertise and the will of the community, and code already present on the Web cannot be exploited.

## 3  Long-term perspectives

Transitioning from open code to Linked Open Code is challenging, yet it represents tremendous opportunities for diverse communities.

Linking data and code in a standard way would open perspectives to fully open and link libraries of programming languages and tools. This promising step enables to auto-construct, from scratch, small utilities computing data. Initiatives like *DeepCode* [7] for code completion can use this work to improve their models. Later on, frameworks such as GPT-3 can be trained on such data. One can also imagine shareable Deep Learning models in the same way, alongside their data, and in a ready-to-use negotiable format. Another important aspect granted by this transition is datasets of cross-language linked functions, ready to use as a base for code translation projects. We believe that syntactical code translation of code artifacts is not enough to achieve the same performance level obtained by experts of each language. Visual Programming Languages (VPLs) started emerging in the last decades and allow users to create programs and algorithms by assembling visual blocks instead of writing actual code. By providing a consistent organization of the information, they allow better performance in design and problem-solving [19] and bring programming to non-specialists. Visual programming environments are not only developed for teaching purposes [8] but also to support the design of real-world applications [20] and workflows such as the Node-RED [9] language, widely use in the context of the Internet of Things. Providing a structure allowing to reuse code assets as black-boxes would allow the emergence of a global VPL to build software relying on functions available on the Linked Open Code.

We think that the FAIR code vision is not FAIR enough when applied to the Semantic Web. Multiple resources openly available on the Web are not used to their full potential.

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
