# OpenReview forum: "Towards a Linked Open Code"
_eswc-conferences.org/ESWC/2021/Conference/Research_Track — ESWC 2021 Research_

### Official Review · AnonReviewer5 · 2021-01-05
**An interesting vision of linked function programming**

**Confidence:** 2
**Impact:** 3
**Design And Technical Quality:** 3

**Review:**

The paper presents the vision of linked open code employing a perspective of openly reusable and FAIR functions annotated with corresponding metadata.
While the vision is captivating, the presented paper does not sufficiently discuss prior related work, real-world challenges to address and practical trade-offs for its realization.
Thus I consider this a borderline paper.

**Anonymity:**

Yes, I would like my review to remain anonymous.

**Rating:**

-1: Weak Reject

**Reuse And Availability:**

3: Medium

**Strong Points:**

1) Captivating vision of FAIR software development.
2) Nice figures, illustrating and summarizing important conceptual aspects.

**Subreviewer:**

I submitted this review.

**Weak Points:**

1) Lacks a discussion of security and scalability implications of such a paradigm, as well as fundamental trade-offs and limitations, such as the shared execution environment needed for diverse functions. (Could e.g. be provided through WebAssembly)
2) The problem which linked open code could solve (and at what cost) could be stated more clearly. Subsequently, it would be clearer whether to categorize this as a `real` or `artificial` problem.
3) Lacks references to and discussion (and possibly awareness) of the related paradigms Named Function Networking (NFN), Function-as-a-Service (FaaS) computing and the prior work done on Semantic Web Service compositioning.
4) Extending upon these shortcomings, an analysis of why this approach has not been implemented so far is missing.

---

> ### Author Rebuttal · Authors · 2021-01-28
>
> We thank the reviewer for their excellent feedback. Hereafter, we give key elements to clarify some aspects of our vision.
>
> The main challenge related to scalability in our vision is to (i) crawl and retrieve information from a large amount of (and ideally all) public open-source repositories, (ii) identify the features implemented by each piece of code, and (iii) store this information. While (i) and (iii) have been tackled recently by other projects dealing with large-scale code crawling and storing (e.g. Software Heritage[1]), (ii) remains an open problem as the use of LSI techniques for feature identification is often restricted to a single software (i.e. homogeneous code conventions) or a single context (i.e. homogeneous domain) and has not yet been applied in the context of our vision (heterogeneous code conventions and domains).
>
> In the NFN paradigm, functions are uniquely identified by a name. A user in the network can call one or several functions by using their names and the execution will be orchestrated by the network. This requires (i) the functions to have a unique implementation accessible on the network and (ii) the user to already know the functions he wants to use and how to use them (input parameters, output type, etc.). In our vision, instead, the same functionalities can be implemented by several functions (which are therefore uniquely identified but semantically linked to each other) and users can query and retrieve the best implementation of the function they want to use.
>
> In the FaaS paradigm, small pieces of code are deployed in the cloud to be queried by each other and called remotely. Although FaaS functions can be made of multiple code functions, we could imagine using FaaS as a part of the implementation of our vision by deploying every code function in a cloud function.
> However, this requires to be able to (i) extract modular code functions, and (ii) identify the functionalities implemented by each code function to semantically link them together. We addressed both points in our vision.
>
> The Semantic Web Service paradigm allows representing a Web Service using an ontology. In our vision, the goal is to generalize this approach to all open-source code repositories available online.
>
> Concerning the already existing approaches, we tried to provide as many examples as we could while remaining within the given page limit. Nevertheless, we will further discuss and add references to other similar approaches in the final version of the paper.
>
> References:
> [1] https://archive.softwareheritage.org/

---

### Official Review · AnonReviewer4 · 2021-01-11
**Work not cosidering relevant literature on the field, most ideas already discussed**

**Confidence:** 5
**Impact:** 1
**Design And Technical Quality:** 2

**Review:**


The paper envisions "Linked Open Code", a Semantic Web approach to share source code (Java).
According to authors' vision, the main challenges to be addressed should be the use of IRIs to reference functions, building an ontology for functions, providing semantic annotations for functions, linking similar functions and finally negotiating functions.
The paper presents some initial reasoning on such an approach and provides initial desiderata and motivations.

Unfortunately the main ideas have been already published and also implemented in relevant work that authors are not considering, such as the so-called CodeOntology project (http://codeontology.org/):

- CodeOntology: RDF-ization of Source Code. M. Atzeni, M. Atzori International Semantic Web Conference (2) 2017: 20-28
- call: A Nucleus for a Web of Open Functions. M. Atzori. International Semantic Web Conference (Posters & Demos) 2014: 17-20
- Toward the Web of Functions: Interoperable Higher-Order Functions in SPARQL. M. Atzori. International Semantic Web Conference (2) 2014: 406-421

and other work in literature such as:

- Implementation-independent function reuse. B De Meester, T Seymoens, A Dimou, R Verborgh. - Future Generation Computer Systems Journal 110: 946-959 (2020)
- Discovering and Using Functions via Semantic Querying. L Noterman, Ghent University, Master Thesis. - 2018
- SoftKG: Building A Software Development Knowledge Graph through Wikipedia Taxonomy". J. Wang, X. Shi, L. Cheng, K. Zhang and Y. Shi. 2020 IEEE World Congress on Services (SERVICES), Beijing, China, 2020.

to mention a few.
The paper should undertake a major review showing original contributions, ideas and vision w.r.t. current state of the art in the field. Anyway, they appear to be insufficient in its current state.


> Post-rebuttal

I acknowledge I've read the rebuttals in your reviews.
What you wrote: "What we propose in our vision is to make use of the existing code in code repositories, providing a "language transparent and efficient" execution (in the sense of "programming language"). We do so by parsing, referencing, linking, and negotiating function code from the pre-existing sources." looks very related to the purposes of the CodeOntology work. For instance, in [7] the authors use CodeOntology to automatically (and transparently) execute code from existing code in code repositories. This is necessarily done by taking into account semantics (also done via DBpedia tags extracted from code comments). As you pointed out, the work focuses mainly on Java although the underlying ontology focus on Object oriented programming languages in general.
I agree that going "Towards a Linked Open Code" is an important vision that deserves publication eventually, but this should be presented with a very clear view of current state of the art, in order to focus on what is missing or what needs to be changed in current direction.
My suggestion is therefore to carefully review existing papers suggested by these reviews, and address them in your vision paper, better clarifying the originality of your view w.r.t. existing work.
I'm mainly concerned with the fact that this may not be fully addressed in the short time available for the camera ready version of the paper, and that there may be still substantial overlap with existing work in literature.

[7] M. Atzeni, M. Atzori: What Is the Cube Root of 27? Question Answering Over CodeOntology. International Semantic Web Conference (1) 2018: 285-300





**Anonymity:**

Yes, I would like my review to remain anonymous.

**Rating:**

-1: Weak Reject

**Reuse And Availability:**

1: Very low

**Strong Points:**


- good presentation, simple to read
- well motivated
- interesting problem

**Subreviewer:**

I submitted this review.

**Weak Points:**


- most ideas (also implementations) already published in literature by other authors
- despite existing solutions have been proposed in the past, only a vision has been discussed without suggesting directions w.r.t. existing work

---

> ### Author Rebuttal · Authors · 2021-01-28
>
> We thank the reviewer for their excellent feedback. We found the additional references very useful. Some of them are closely related to the vision we introduce in the paper, but they present some major differences.
>
> For instance, CodeOntology[1] introduces a parser and an ontology to represent and describe Java code structurally, while our vision goes beyond that: it involves parsing and analyzing source code in any language and describing, not only its structure (input/output parameters, dependency, classes, etc..) but also its semantics (implemented functionalities). Moreover, unlike [2] and [3] focusing on remotely executing SPARQL functions, we aim to enable users to discover, download, and locally execute the best implementation of a given functionality in the required programming language.
>
> The paradigm presented in [4] and [5] is very similar to the one we envision, with the main difference (that is also one of the main challenges of our approach) that we aim to automatically discover, identify and annotate the source code, while previous works foresee the manual publication of description and implementations by developers.
>
> In general, we think that these works can constitute a solid basis on which we can rely to start building a first working prototype implementing a part of our vision, for example by extending the ontologies presented and used in [1] and [4] to fit our larger representation needs. We aim to include them in the final version of our paper.
>
> As for [6], the work presented in this paper introduces a knowledge graph describing knowledge about software and not software itself, therefore although it is an interesting work, its scope is beyond the one of our vision.
>
> [1] CodeOntology: RDF-ization of Source Code. M. Atzeni, M. Atzori International Semantic Web Conference (2) 2017: 20-28
>
> [2] call: A Nucleus for a Web of Open Functions. M. Atzori. International Semantic Web Conference (Posters & Demos) 2014: 17-20
>
> [3] Toward the Web of Functions: Interoperable Higher-Order Functions in SPARQL. M. Atzori. International Semantic Web Conference (2) 2014: 406-421
>
> [4] Implementation-independent function reuse. B De Meester, T Seymoens, A Dimou, R Verborgh. - Future Generation Computer Systems Journal 110: 946-959 (2020)
>
> [5] Discovering and Using Functions via Semantic Querying. L Noterman, Ghent University, Master Thesis. - 2018
>
> [6] SoftKG: Building A Software Development Knowledge Graph through Wikipedia Taxonomy". J. Wang, X. Shi, L. Cheng, K. Zhang and Y. Shi. 2020 IEEE World Congress on Services (SERVICES), Beijing, China, 2020.

---

### Official Review · AnonReviewer1 · 2021-01-12
**A vision for a Web in which code in general could be accessed and used as Linked Data: an interesting proposal.**

**Rating:** 1
**Confidence:** 3
**Impact:** 3
**Design And Technical Quality:** 3

**Review:**

As a vision paper, it presents a potentially useful paradigm towards handling "code as data", i.e., as first-class citizens in the Semantic Web information space.  The described challenges cover a general and first approach to achieve a "first working prototype".

Still, some topics are not addressed such as:
- code composition: a higher level of abstraction about organizing functions/procedures (classes, modules/packages, etc.).
- code execution environment: underlying computing resources/architecture about the code.  This topic is slightly presented in the last lines of the "Ranking functions" subsection (pag. 6).
- code licensing.

Specific comments about the paper are presented below:
```{```
* Major:
	- In section 2.1, there is no mention of the *Wikifunctions* project [1][2].  It's my opinion that this project is strongly related to some parts of the proposed vision in this paper.
		+ [1] https://meta.wikimedia.org/wiki/Abstract_Wikipedia
		+ [2] https://www.mediawiki.org/wiki/Extension:WikiLambda
		+ The authors should describe how its related (or not) to the "Linked Open Code" vision.

	- In section 2.2, the paper states: "The ideal is to have a function repository...".  Do the authors imply to have a centralized repo (unique and global) even though the World Wide Web principles define a decentralized information space? How about a Web of decentralized function repos? Your thoughts...

	- One important challenge that is not discussed, it's the one related to code licensing.


* Minor corrections:
	- pag. 1: "ways to store metadata about data"; it should be just "ways to store metadata" (we know what metadata is).
	- pag. 5: "AST analysis" --> write "Abstract Syntax Tree (AST) analysis" for clarity.
	- pag. 6: "Ranking functions" is not part of the challenges depicted in Fig. 1? It should be included in Fig. 1.
```}```

```
* Main evaluation (ranging from 0-100):
	+ Quality: 80
	+ Originality: 90
	+ Significance: 80
```


---
# Post-rebuttal

- I acknowledge I've read the rebuttals in your reviews.
- Just one clarification regarding "Wikifunctions": actually, this project is aiming to build a general repo of functions (for any programming language) and not only to use for Wikidata.
- My final evaluation of the paper stands.


**Anonymity:**

Yes, I would like my review to remain anonymous.

**Reuse And Availability:**

1: Very low

**Strong Points:**

It's a useful vision that could have a great impact on the coding landscape over the Web.

**Subreviewer:**

I submitted this review.

**Weak Points:**

+ One important challenge that is not discussed, it's the one related to *code licensing*.  Any thoughts about it?
+ Other important topics that should be addressed are:
	+ #1: code composition: include higher levels of abstraction about organizing functions/procedures (classes, modules/packages, namespaces, etc.).
	+ #2: code execution environment: underlying computing resources/architecture about the code, and handling of "data privacy" for execution/testing purposes (see below).
+ Regarding #2: if users would share the execution log as mentioned in "Ranking functions" (pag. 6), how to address the privacy of the used dataset if the dataset is deemed "private"?

---

> ### Author Rebuttal · Authors · 2021-01-28
>
> We thank the reviewer for their excellent feedback. We answer some specific questions below, and will incorporate all feedback in the manuscript.
>
> 1. "In section 2.1, there is no mention of the Wikifunctions project"
> Wikifunctions helps in creating natural language-independent articles based on Wikidata. Unlike that, our vision aims to make use of the existing code in code repositories and provide a "language transparent" execution (in the sense of "programming language"). This can be achieved by parsing, referencing, linking, and negotiating function code from the pre-existing sources. Wikifunction is proposing a specific translation use-case, unlike our vision which aims to semantically link code in open-source code repositories with linked data.
>
> 2. "Do the authors imply to have a centralized repo (unique and global) even though the World Wide Web principles define a decentralized information space? How about a Web of decentralized function repos? Your thoughts..."
> A centralized approach is certainly a first step (as is the case for The Function Hub[1] for example). However, in a long-term vision, developers publishing and storing an RDF version of their code (automatically generated by a framework) could be a more viable solution, compliant with the World Wide Web standards. This way, storage will be distributed but this solution opens a new challenge about distributed querying.
>
> 3. "One important challenge that is not discussed, it's the one related to code licensing. Any thoughts about it?"
> Information related to code licensing is important as licensing imposes constraints on the reuse of software. Therefore, this information must be added to the description of data in the ontology to be presented to the user making use of these functions. We aim to address such an issue in future productions.
>
> 4. "code composition: include higher levels of abstraction about organizing functions/procedures (classes, modules/packages, namespaces, etc.)."
> In our vision, every single code unit (regardless of its granularity) is parsed, identified, and described as such in the Linked Data representation. For example, methods would be related by semantic links to the class implementing them, and the same for classes with the package they belong to. The ontology we took as an example describes how functions are linked to packages and dependencies.
>
> 5. "code execution environment: underlying computing resources/architecture about the code, and handling of "data privacy" for execution/testing purposes" and "if users would share the execution log as mentioned in "Ranking functions" (pg. 6), how to address the privacy of the used dataset if the dataset is deemed "private"?"
> In our vision, execution logs provided by users are a way to improve the ranking of functions. Users are asked to give their consent to share them, the same way current applications ask users if they accept to share anonymous data to improve the service. However, users would need to share data necessary for our vision (e.g. information on software architecture and installed libraries).
>
>
> References:
> [1] The Function Hub: An Implementation-Independent Read/Write Function Description Repository. ESWC (Satellite Events) 2019: 33-37

---

### Official Review · AnonReviewer2 · 2021-01-14
**Significant overlap with existing work**

**Confidence:** 4
**Impact:** 1
**Design And Technical Quality:** 3

**Review:**

This is a "Problems to Solve Before You Die" paper with the problem statement "how to take advantage of code as a pre-existing structured, and functional type of data in the      Semantic Web". The problem has merit, unfortunately, the authors seem to be unaware of      the work by Ben De Meester and colleagues which significantly overlaps with the proposed problem. This work is      reported in several publications:

- Implementation-independent function reuse. Future Gener. Comput. Syst. 110: 946-959 (2020)
- The Function Hub: An Implementation-Independent Read/Write Function Description Repository. ESWC (Satellite Events) 2019:        33-37
- An Ontology to Semantically Declare and Describe Functions. ESWC (Satellite Events) 2016: 46-49
- Discovering and Using Functions via Content Negotiation. International Semantic Web Conference (Posters &        Demos) 2016

For this reason it is not possible to accept the paper.

The call for such papers asks for /an argument why this problem      should be solved/. Although a list of possibilities for      exploiting a possible solution to the problem are given, I find      the suggestions to be too concrete; I miss stronger and more      convincing, and visionary arguments for why the proposed problem      is "hard, longstanding or paradigm-breaking" (quote from the      call for papers)

**Anonymity:**

Yes, I would like my review to remain anonymous.

**Rating:**

-2: Reject

**Reuse And Availability:**

3: Medium

**Subreviewer:**

I submitted this review.

---

> ### Author Rebuttal · Authors · 2021-01-28
>
>
> We thank the reviewer for their excellent feedback.
>
> We believe that the work by Ben De Meester et al. is in the same direction, and covers many points we wanted to address according to their specific use-case. The mentioned works propose linking code from NPM packages, Javascript, and Java. We agree that such an overlapping work should have been cited. However, our focus was centered on the semantic interlinking of functions and in the mechanism for automatic code discovery in a "language-independent" way. Instead of taking the "fno:" ontology for our unique example due to the imposed page limit, we proposed the R function ontology that we thought would be closer to our vision.
>
> We are sorry to hear that our presentation lacks more visionary arguments. The vision we mentioned was not meant only for linking code, but to shift the paradigm of open-source code into a semantically-linked one allowing agents to negotiate, auto-translate, reuse, and find the suitable executable code version for their use-case. Due to the limited space, we couldn't address in detail different issues such as code licensing, version control, features of code semantics, and other topics related to our vision.
>
> We are happy to include the previous works in our pipeline to clarify the overlap and the differences.

---

### Official Review · AnonReviewer3 · 2021-01-17
**Nice vision paper, but ignores important work towards the proposed vision**

**Rating:** 1
**Confidence:** 5
**Impact:** 4
**Design And Technical Quality:** 4

**Review:**

I think that this is an interesting vision paper, which addresses work very close to my research interests, and of the interests of the Semantic Web community at ESWC. Therefore I think it would be a nice contribution to the conference. In fact, I would be happy to discuss more about this work with the authors.

That said, I think there is much ongoing work that has to started to enable the vision proposed in the paper. For example, the  Function Ontology (https://fno.io/spec/) already proposes a simple representation for functions according to semantic web standards; and the recent Abstract Wikipedia and wikifunctions (https://meta.wikimedia.org/wiki/Abstract_Wikipedia) provides an initial implementation of a representation for executable functions with an ambitious scope (even if no negotiation is involved).

The level of granularity at which functions are described is quite important. For example, in my own work we had to propose ontologies of software components to represent models in environmental sciences (https://w3id.org/okn/o/sd/). This goes beyond parameter representation, as the inputs are usually files which contain variables of very different nature.

In the scientific workflow domain, there have been for years catalogs of web services, workflows and components which are semantically described in terms of inputs and outputs (see Gil et al 2011, https://www.isi.edu/~gil/papers/gil-etal-ieee-is-11.pdf). These are functions at a higher level of abstraction, but functions nonetheless. Probably the biggest challenge is not the semantic annotation of input/output parameters or content negotiation, but linking functions that are similar in scope. That said, I really liked the proposal to do content negotiation based on the desired implementation of a function


**Anonymity:**

No, I would like my review to be deanonymized.

**Reuse And Availability:**

3: Medium

**Strong Points:**

- Nice combination of semantics and code, I agree with the intent of the authors.

**Subreviewer:**

I submitted this review.

**Weak Points:**

- Several approaches (specifically abstract Wikidata) work already towards the vision proposed here

---

> ### Author Rebuttal · Authors · 2021-01-28
>
>
> We thank the reviewer for their excellent feedback.
>
> To address the two points highlighted by the reviewer:
> - The Function ontology: we believe that the "fno:" ontology offers a good representation for functions in their general use-case (linking the functions to algorithms). However, we preferred to mention the R function ontology as our only example due to the page limit, as for us it was closer to the real use-case we were working on and included the parts we addressed such as parameter referencing, execution workflow, and dependency. Despite missing the part about linking functions semantically, we believed the R function ontology was closer than the "fno:". However, we would be happy to add that information and compare the limits of both before moving forward to the pipeline.
>
> - Abstract Wikidata: the "Language-independent" aspect of Wikifunctions is not similar to our vision.
> Wikifunctions helps in creating natural language-independent articles based on Wikidata. What we propose in our vision is to make use of the existing code in code repositories, providing a "language transparent and efficient" execution (in the sense of "programming language"). We do so by parsing, referencing, linking, and negotiating function code from the pre-existing sources. Wikifunction is proposing a specific translation use-case, unlike our vision aiming to semantically link code in open-source code repositories with linked data.
>
> We are grateful to the reviewer for their feedback, and we will be more than happy to discuss further opportunities to pursue this work.

---

> > ### Comment · AnonReviewer3 · 2021-01-28
> > **Please do not dismiss important related work**
> >
> > I think the authors are dismissing wikifunctions as not relevant, but I quote: "Wikifunctions is a collaboratively edited catalog of functions that aims to allow the creation, modification and reuse of code" (https://en.wikipedia.org/wiki/Wikifunctions). They have an initial catalog of functions which are ongoing work: https://meta.wikimedia.org/wiki/Abstract_Wikipedia/Early_function_examples (I believe some of them have implementations), and an architecture proposal on how would the whole system work.
> >
> > Wikifunctions is related to abstract Wikipedia, which has a general goal of translation. However the goal of Wikifunctions is very similar to what is envisioned by the authors.
> >
> > The other work I pasted, not addressed in the rebuttal, addresses other important aspects of the semantics of code and software functionality.

---

### Decision · Program_Chairs · 2021-02-23

**Decision:**

Accept with shepherding

**Comment:**

The reviewers agree that the paper addresses an interesting and important topic. However, existing work in that field has not been taken into account and sufficiently discussed, as e.g. Wikifunctions, Function Ontology, or CodeOntology. Therefore, the paper will be only accepted with shepherding if it fulfills the following requirements:

(1) Take into account already existing work as mentioned by the reviewers.

(2) Discuss how your problem/vision differs from the work already done or going on.

(3) Further take into account code composition, code execution, and code licensing.

(4) Discuss security and scalability implications of your approach.